# Vascularization and biocompatibility of poly(ε-caprolactone) fiber mats for rotator cuff tear repair

Sarah Gniesmer[1,2], Ralph Brehm[3], Andrea Hoffmann[2,4], Dominik de Cassan[5], Henning Menzel[5], Anna Lena Hoheisel[2,6], Birgit Glasmacher[2,6], Elmar Willbold[2,7], Janin Reifenrath[2,7], Nils Ludwig[8], Ruediger Zimmerer[1], Frank Tavassol[1], Nils-Claudius Gellrich[1], Andreas Kampmann[1,2]*

1 Department of Oral and Maxillofacial Surgery, Hannover Medical School, Hannover, Germany, 2 NIFE—Lower Saxony Centre for Biomedical Engineering, Implant Research and Development, Hannover, Germany, 3 Institute for Anatomy, University of Veterinary Medicine Hannover, Hannover, Germany, 4 Department of Orthopedic Surgery, Laboratory for Biomechanics and Biomaterials, Graded Implants and Regenerative Strategies, Hannover Medical School, Hannover, Germany, 5 Institute for Technical Chemistry, Braunschweig University of Technology, Braunschweig, Germany, 6 Institute of Multiphase Processes, Leibniz University Hannover, Hannover, Germany, 7 Department of Orthopedic Surgery, Hannover Medical School, Hannover, Germany, 8 Department of Pathology, University of Pittsburgh School of Medicine, Pittsburgh, PA, United States of America

* Kampmann.Andreas@mh-hannover.de

**Data Availability Statement:** All relevant data are within the manuscript.

**Funding:** This research project has been supported by the German Research foundation (DFG),

## Abstract

Rotator cuff tear is the most frequent tendon injury in the adult population. Despite current improvements in surgical techniques and the development of grafts, failure rates following tendon reconstruction remain high. New therapies, which aim to restore the topology and functionality of the interface between muscle, tendon and bone, are essentially required. One of the key factors for a successful incorporation of tissue engineered constructs is a rapid ingrowth of cells and tissues, which is dependent on a fast vascularization. The dorsal skinfold chamber model in female BALB/cJZtm mice allows the observation of microhemo-dynamic parameters in repeated measurements in vivo and therefore the description of the vascularization of different implant materials. In order to promote vascularization of implant material, we compared a porous polymer patch (a commercially available porous polyure-thane based scaffold from Biomerix™) with electrospun polycaprolactone (PCL) fiber mats and chitosan-graft-PCL coated electrospun PCL (CS-g-PCL) fiber mats in vivo. Using intra-vital fluorescence microscopy microcirculatory parameters were analyzed repetitively over 14 days. Vascularization was significantly increased in CS-g-PCL fiber mats at day 14 compared to the porous polymer patch and uncoated PCL fiber mats. Furthermore CS-g-PCL fiber mats showed also a reduced activation of immune cells. Clinically, these are important findings as they indicate that the CS-g-PCL improves the formation of vascularized tissue and the ingrowth of cells into electrospun PCL scaffolds. Especially the combination of enhanced vascularization and the reduction in immune cell activation at the later time points of our study points to an improved clinical outcome after rotator cuff tear repair.

research unit FOR 2180 "Gradierte Implantate für Sehnen-Knochen-Verbindungen". Grant numbers KA 4236/1-1 and KA 4236/1-2 to AK. The funders had no role in study design, data collection and analysis, decision to publish, or preparation of the manuscript.

**Competing interests:** The authors have declared that no competing interests exist.

## Introduction

Rotator cuff tears are common injuries in human shoulders. Especially degenerative processes in elderly people account for a large number of injuries, so that about 50% of people over their seventies are affected by a rotator cuff tear [1]. Depending on the extent of the tear, the symptoms include pain, reduced function and weakness. Especially lateral abduction of the affected arm causes severe pain, which persists even throughout the night while resting [2]. Frequently, changes in the musculotendinous unit occur between the onset of the rotator cuff tear and the time of diagnosis. The loss of muscle activity thereby leads to muscle atrophy and the loss of pretension entails fatty infiltration of the musculature [3,4]. As a result tendon retraction arises, as well as loss of elasticity in the musculotendinous unit [5]. These changes complicate or in some cases hinder surgical fixation of the tendon to the osseous attachment. The main problem after surgery is the healing process after tendon reconstruction, because instead of a regenerated tendon-bone-junction [6] the formation of scar tissues, which show reduced tensile strength compared to the intact tendon, can be observed [7]. For regeneration of tendon defects, different graft materials and surgical techniques were developed to support tendon healing [6], but neither special techniques nor different graft materials are able to improve the prognosis after tendon reconstruction significantly. Furthermore, also after successful refixation a reduced tendon strength compared to healthy tendons can be observed [8] and failure rates after tendon reconstruction remain unacceptably high. The high incidence of failure related to existing implant materials and repair techniques emphasize the importance of functional solutions for tendon repair [9].

Implants that facilitate an effective surgical treatment of rotator cuff tears ideally mimic the biological and biomechanical characteristics of the intact tendon and at the same time do not hinder the natural healing process. A key prerequisite for a successful implant material is the rapid ingrowth of cells and tissues for the development of a regenerated tendon-bone-transition. Ingrowth of cells and tissues thereby strongly depends on a fast vascularization of the implant material for transporting nutrients, growth factors, and supporting gas exchange and removal of waste materials. Implant failure is often attributable to insufficient tendon healing, mainly depending on insufficient vascularization [10,11]. Potential candidates to fulfill these criteria are electrospun fiber mats, which are already often used in tissue engineering applications [12]. Polycaprolactone (PCL) is applicable in this field because of its adequate mechanical properties, its biocompatibility and slow degradation in vivo [13]. However, its hydrophobic nature interferes with cell attachment and growth [14]. Recently it was shown that electrospun PCL fiber mats with a CS-g-PCL coating are more hydrophilic and provide cellular recognition sites, which create an attractive surface for initial cell attachment [15,16].

In recent decades biodegradable polymers, like PCL, have become increasingly important. PCL has a wide range of application in tissue engineering including vascular grafts [17], bone [18,19], cartilage [20], liver [21], bladder [22], skin [23] and nerve [24]. Implants for rotator cuff tear repair should have a positive impact on tendon healing by supporting cellular infiltration and guiding the regeneration of an organized tendon structure. These positive effects could be shown for nanofibrous PCL based scaffolds in an *in vivo* model in rats for primary rotator cuff repair [25]. Electrospun PCL fiber mats are particularly suited as an implant for rotator cuff tear repair as they can easily be produced with variable fiber arrangements. By fabrication of fiber mats with directed or undirected fiber orientation they can be adopted to the microarchitecture of the different sections of the native tendon. The specific design of the fiber mat in turn stimulates the regeneration of organized tendon structures. In a previous study we analyzed electrospun PCL fiber mats with undirected fiber orientation, that were intended to mimic the specific fiber orientation at the tendon-bone transition in the femur chamber in rats

[26]. In this study fiber mats with directed fiber structures were used to simulate the tendon-muscle transition. These electrospun PCL fiber mats are intended to form the lead structure for the transition zone of the rotator cuff.

Different studies investigated the physical, chemical and biological characteristics of scaffold devices for rotator cuff repair [27], but data describing the vascularization of electrospun PCL fiber mats are limited. For characterization of the early vascularization of electrospun PCL implant materials the dorsal skinfold chamber in mice was used in the present study. This in vivo model allows the repetitive observation of the vascularization of different implant materials by means of intravital fluorescence microscopy [28]. The aim of this study was to compare microhemodynamic parameters inside electrospun PCL based implants as described above with the microhemodynamic parameters of a reticulated polyurethane based scaffold (Biomerix™, Biomerix Corporation, Fremont, CA), which was used in studies for rotator cuff repair before [29].

## Materials and methods

### Implants

Electrospun polycaprolactone fiber mats with aligned fibers were produced as described elsewhere [30]. Briefly, a poly-ε-caprolactone solution of 170 mg/ml ($M_n$ = 80.000 g/mol, Sigma-Aldrich Chemie GmbH, Taufkirchen, Germany) (PCL) and trifluoroethanol (TFE, abcr GmbH, Karlsruhe, Germany) was used to produce electrospun PCL fiber mats. The scaffolds were produced with a collector speed of 8 m/s, which resulted in orientated fibers. The voltage was set to 25 kV and the emitter to collector distance to 25 cm. By using a flow rate of 4 ml/h 8 ml polymer solution resulting in fiber mats with a thickness of approximately 200 μm.

Experimental groups were unmodified PCL fiber mats and PCL fiber mats modified with a fiber coating utilizing a graft copolymer consisting of chitosan and polycaprolactone, called CS-g-PCL [31]. As a control group a commercially available porous polyurethane based scaffold (Biomerix RCR Patch, Biomerix Corporation, Somerset, USA; purchased from Cellon, Bascharage, Luxembourg), was used [29,32].

For sterilization all implants were cut into pieces of 3 mm x 5 mm and shrink-wrapped in sterilization pouches (SteriClin, Vereinigte Papierwarenfabriken, Feuchtwangen, Germany). Sterilization was performed by beta radiation with a dose of 25 kGy (using a Rhodotron TT 100 e-beam accelerator, Mediscan, Kremsmünster, Austria).

### Evaluation of porosity and pore diameter

Gravimetric method was used to determine the porosity of the fibrous scaffolds by using the mass, density and dimensions of a sample. The pore diameter was determined with a capillary flow porometer (3 Gzh, Quantachrome GmbH & Co.KG). This characterization technique based on the displacement of a wetting liquid (Porofil®, Quantachrome GmbH & Co. KG) from the pores of the sample by increasing the pressure from an inert gas. The underlying equations for the calculation of the porosity and the pore diameter were described elsewhere [26]. The porosity and the pore diameter of the reticulated polyurethane based scaffold was evaluated in a previous study [26].

### Experimental protocol

For intravital fluorescence microscopy, a total of 24 BALB/cJZtm mice were equipped with dorsal skinfold chambers. Scaffolds, consisting of either porous polymer patch (control) (n = 8) or unmodified PCL fiber mats (n = 7) or chitosan coated PCL fiber mats (n = 9), were

implanted into the dorsal skinfold chamber. Intravital fluorescence microscopy analysis of volumetric blood flow, wall shear rate, leukocyte-endothelial cell interaction, macromolecular leakage and functional capillary density were performed immediately as well as 3, 6, 10 and 14 days after implantation.

## Animals

All experiments were conducted in accordance with the German legislation for the protection of animals and the Guide for the Care and Use of Laboratory Animals (8[th] edition, 2011). The experiments were approved by the competent authority (Niedersächsisches Landesamt für Verbraucherschutz und Lebensmittelsicherheit, reference number 33.12-42502-04-15/2015). Female BALB/cJZtm mice (Central animal facility, Hannover Medical School) with an age of 12 to 18 weeks and a body weight of 22.5 g ± 1.9 g were used for the study. All animals were housed individually per cage at room temperature between 22˚C and 24˚C and a relative humidity of 60–65% with a 12-hour day-night cycle. The mice had free access to tap water and standard pellet food (1328 Hybridpellet, Altromin, Lage, Germany) at all times.

## Anesthesia

The preparation of the dorsal skinfold chamber and repeated intravital fluorescence microscopy were executed under inhalational anesthesia (EZ-7000 Classic System, PLEXX, Elst, The Netherlands) by isoflurane (Isofluran CP®, cp pharma, Burgdorf, Germany). After the animals were placed in an induction chamber anesthesia was induced with 5% isoflurane until loss of righting reflex. The anesthesia depth was determined with the aid of the toe pinch. After induction the animals were placed in appropriate position on a heating mat to protect them against hypothermia. Animals remained anesthetized with the aid of a nose cone with 2–3% isoflurane in oxygen.

## Preparation of the dorsal skinfold chamber

The dorsal skinfold chamber in mice is an accepted model to investigate changes of the microcirculation and the biocompatibility of a given material [33]. The chamber is constructed of two symmetrical frames and has an observation window with 12 mm diameter (Fig 1). The preparation of the dorsal skinfold chamber has been described in detail previously [34,35]. Briefly, after preparation of the skin the frames were implanted on the extended dorsal skinfold of the mice, so that the skin was double-layered between the frames. One layer of skin including cutis, subcutis, musculus panniculus carnosus, and the two layers of the musculus retractor was completely removed by dissection in a circular area with a diameter of 12 mm, exposing the musculus panniculus carnosus of the opposite skin layer. The exposed tissue layer was covered with a coverslip that was secured with a circlip. Postoperatively, the animals were given three days for convalescence and adaption to the dorsal skinfold chamber before the implant material was tested *in vivo*. For implantation the coverslip was removed and the implant material was inserted in the center of the observation window. The mice were treated with antibiotics (Enrofloxacin 5 mg/kg, Baytril 25 mg/ml, Bayer, Leverkusen, Germany) intraoperatively and per os for seven days after surgery. Analgesics (Carprofen, 5 mg/kg, Rimadyl 50 mg/ml, Pfizer Deutschland GmbH, Berlin, Germany) were given intraoperatively and for two days after surgery. During the experimental period the animals were evaluated daily with a score based evaluation system including the assessment of weight loss, general condition, spontaneous behavior, clinical findings and lameness.

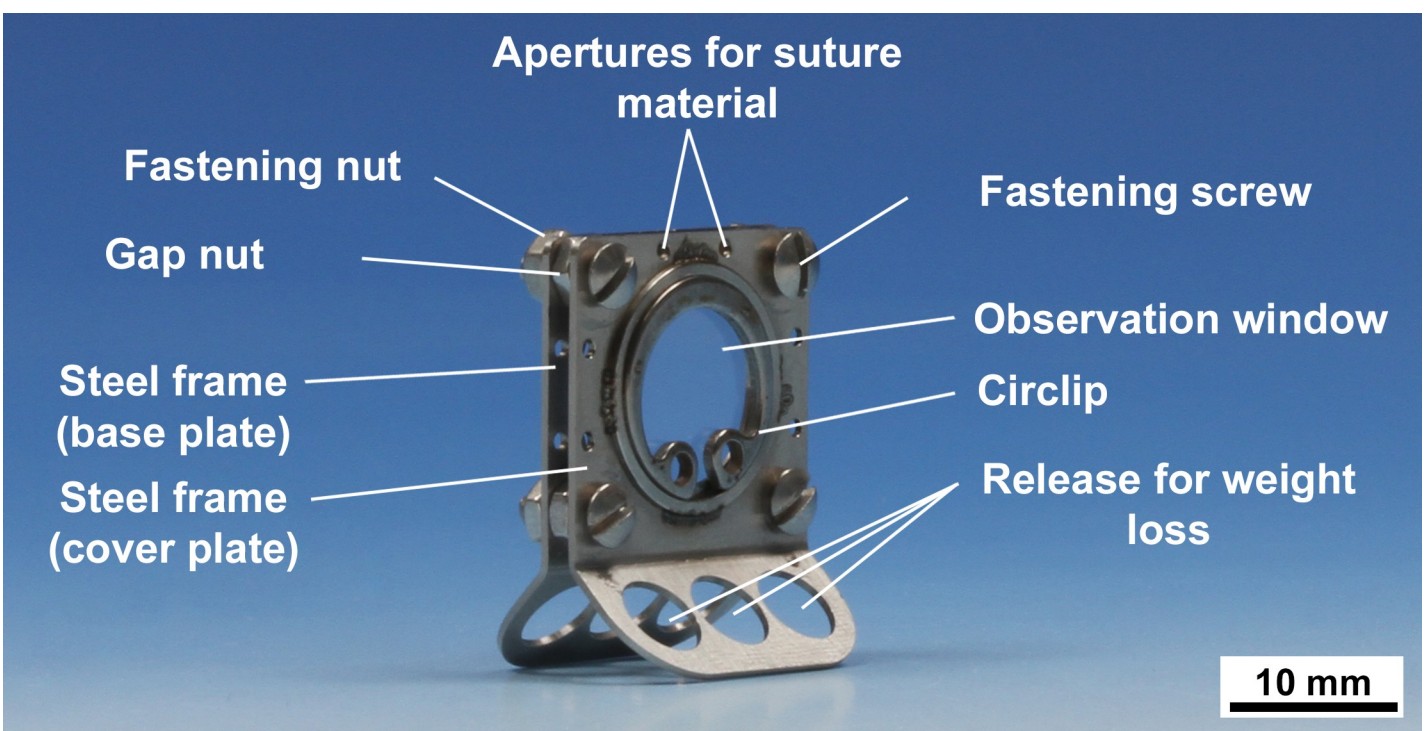

**Fig 1. Dorsal skinfold chamber.** Two symmetrical steel frames are connected via screws with a gap nut in between. The observation window (Ø 12 mm) is covered with a coverslip, which is fixed by a circlip. Recesses reduce the weight of the chamber.

### Intravital fluorescence microscopy and analysis of microcirculatory parameters

Intravital fluorescence microscopy was performed as described in Gniesmer et al., 2019 [26]. The recorded data was analyzed using the image analysis software CapImage (CapImage 8.6.3, Zeintl, Heidelberg, Germany). Briefly leukocyte-endothelial cell interactions, microhemodynamics, and macromolecular leakage were measured in four different regions of interest in the border zone of the scaffold. In each region of interest one venule in the granulation tissue around the implant (inner diameter: 20–40 μm) was selected and observed over 20 seconds for evaluation of vessel diameter, red blood cell velocity, wall shear rate and macromolecular leakage. The leukocytes were classified according to their interaction with the vascular endothelium as adherent (they that did not move or detach from the endothelial lining within an observation period of 20 s), rolling (moving cells with a velocity less than two-fifth of the centerline velocity) or free flowing cells. Microvessel density, defined as the length of blood vessels per area of observation given in cm/cm$^2$, was measured in the periphery around the implant and in the center of the implant. Both values were displayed as the total functional capillary density expressed as the sum of peripheral and central functional capillary density. For a full description of the data analysis refer to Gniesmer et al., 2019 [26].

### Histology and immunohistochemistry

After the experimental period of 14 days, animals were finalized via cervical dislocation under deep anesthesia and histological examinations were performed. Formalin-fixed specimens of the dorsal skinfold chamber were embedded in paraffin for light microscopy according to standard procedures. Thin sections (5 μm) were stained with hematoxylin (Merck KGaA,

Darmstadt, Germany) and eosin (Merck KGaA, Darmstadt, Germany) according to standard procedures and examined by light microscopy (Leica DM4000 B, Leica Mikrosysteme, Wetzlar, Germany).

Van Gieson staining of thin sections (5 μm) with hematoxylin and picrofuchsia acid solution (Merck KGaA, Darmstadt, Germany) according to standard procedures was applied to detect collagen fibers. All specimens were examined by light microscopy (Leica DM4000 B, Leica Mikrosysteme, Wetzlar, Germany).

For detection of capillaries, endothelial cells were immunohistochemically stained using a rabbit anti-mouse CD31 antibody (LifeSpan Biosciences, Seattle, USA purchased from BIO-ZOL Diagnostica Vertrieb GmbH, Eching, Germany); a rat anti-mouse CD68 antibody (Acris Antibodies GmbH, Herford, Germany) was used to detect macrophages; a rabbit anti-mouse CSF1R antibody (OriGene Technologies, Inc., Rockville, USA) was used for detection of macrophages in general; a rat anti-mouse CD86 antibody (OriGene Technologies, Inc., Rockville, USA) was used to detect macrophages with predominantly pro-inflammatory actions; a rabbit anti-mouse CD11b antibody (OriGene Technologies, Inc., Rockville, USA) was used to detect monocytes and a rat anti-mouse CD3 antibody (OriGene Technologies, Rockville, Inc., USA) was used to detect T-cells.

As secondary antibodies a biotin conjugated goat anti-rabbit antibody and a biotin conjugated goat anti-rat antibody (both from Dianova, Hamburg, Germany), respectively were used. After incubation with streptavidin conjugated horseradish peroxidase (Dianova, Hamburg, Germany) color development with the addition of 3.3´-diaminobenzidine (DAB) (Vector Laboratories, Inc., Burlingame, CA, USA) was monitored microscopically followed by counterstaining with hematoxylin. For immunofluorescence detection an Alexa Fluor® 488-conjugated goat anti-rat antibody and an Alexa Fluor® 488-conjugated goat anti-rabbit antibody (both Dianova, Hamburg, Germany), respectively was used. Nuclei were stained using DAPI (Carl Roth GmbH, Karlsruhe, Germany). By omitting the primary antibody negative controls were performed, which all showed no detectable staining. All specimens were examined by fluorescence microscopy (Leica DM4000 B, Leica Mikrosysteme, Wetzlar, Germany).

The software celISens Dimensions 1.14 (Olympus Deutschland GmbH, Hamburg, Germany) was used to quantify the immunofluorescence detection of macrophages, monocytes and T-cells. Using a magnification of 20x the area of green fluorescence was measured and normalized using the area of blue fluorescence (nuclei).

Neutrophilic granulocytes were detected by naphtol-AS-D-chloroacetate esterase staining as described in Willbold et al., 2013 [36].

## Statistics

Results are expressed as means ± standard error of the mean (SEM). Differences between groups were assessed by one-way analysis of variance (ANOVA) and differences within groups were analyzed by one-way repeated measures ANOVA. To identify differences between pairs of groups, Student-Newman-Keuls post-hoc tests were performed. Differences were considered significant at $p < 0.05$. All data were analyzed using the GraphPad Prism software (Version 7.0).

## Results

### Properties of the material

To compare and evaluate them towards the desired biological performance the porosity and the pore diameter of the various implant materials were examined (Fig 2). The control

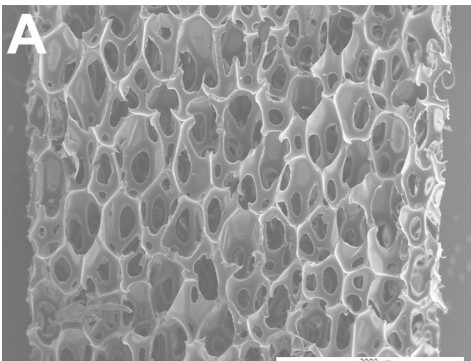
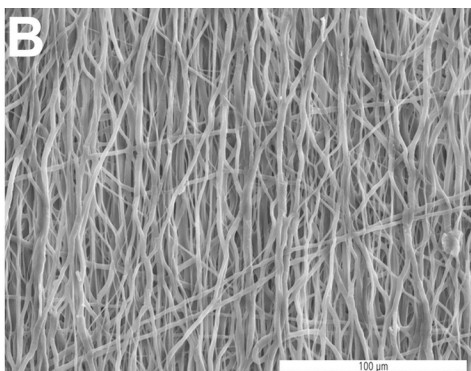
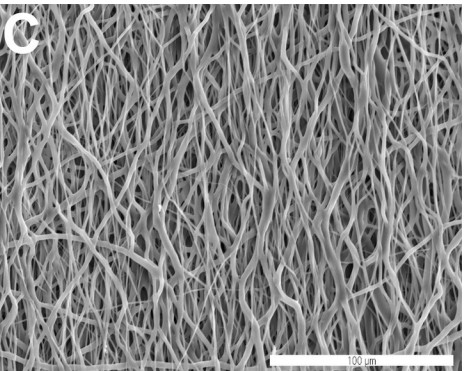

**Fig 2. Scanning electron microscopy of the implant materials.** Porous polymer patch (**A**), unmodified PCL fiber mat (**B**) and PCL fiber mat modified with CS-g-PCL (**C**). Scale bars: 200 μm (A), 100 μm (B, C).

polyurethane material exhibited very large pore sizes of up to 175.34 μm and a porosity of 94.7% [26]. It was found that the pure PCL had a porosity of 93.67%. The pore size (maximum diameter 11.5 μm) was significantly lower than for the control group. Coating with the graft-copolymer CS-g-PCL reduced the porosity to 81.79% and simultaneously increased the fiber diameter by 0.45 μm, resulting in a maximum pore diameter of 7.56 μm (Table 1).

## Animals

The animals tolerated the chamber well and showed no signs of discomfort or changes in behavior. The control group and the group with CS-g-PCL lost three animals each. The reasons for the losses were very different. In the control group, one animal each was lost due to anesthesia complications, through weight loss as well as through an injury in cage. In the CS-g-PCL group, there were losses within the anesthesia, due to an inflammation at the chamber as well as due to a necrosis of tail tip. There were no losses in the group with PCL. The lost animals were not included in the statistical evaluation.

## Microhemodynamic parameters

Overall, data indicated that the microvascular blood flow of host tissue was not impaired upon implantation of the different materials. Venular diameters stayed constant in all experimental groups ranging between 19 and 26 μm (Table 2). Volumetric blood flow and wall shear rates of venules were not significantly altered (Table 2).

## Functional capillary density

Within the first three days after implantation, all implants provoked an angiogenic response of the host striated muscle tissue. Quantitative analysis revealed a progressive increase in the

**Table 1. Fiber diameter in μm, porosity in %, the maximum pore diameter in μm, the middle pore diameter in μm and the smallest pore diameter in μm of the control group scaffold (control; see Gniesmer et al., 2019), the unmodified PCL fiber mat group (PCL) and the PCL fiber mat modified with CS-g-PCL group.**

|  | control | PCL | CS-g-PCL |
|---|---|---|---|
| fiber diameter | - | 1.6 μm ± 0.7 μm | 2.1 μm ± 0.6 μm |
| porosity | 94.7% | 93.7% | 81.8% |
| Maximum pore diameter | 175.3 μm | 11.5 μm | 7.6 μm |
| Middle pore diameter | 139.4 μm | 7.5 μm | 5.7 μm |
| Smallest pore diameter | 78.1 μm | 6.1 μm | 4.0 μm |

**Table 2. Venular diameters in μm, volumetric blood flow in pl/s and wall shear rate in s$^{-1}$ of postcapillary and collecting venules at the border zones of the control group scaffold (control), the unmodified PCL fiber mat group (PCL) and PCL fiber mat modified with CS-g-PCL group immediately (d0) and 3, 6, 10 and 14 days after implantation.** Values are expressed as means ± SEM.

| | Day 0 | Day 3 | Day 6 | Day 10 | Day 14 |
|---|---|---|---|---|---|
| *Diameter [μm]* | | | | | |
| control | 24.0 ± 1.1 | 25.3 ± 1.2 | 24.7 ± 4.3 | 21.8 ± 3.7 | 26.0 ± 2.3 |
| PCL | 21.7 ± 1.5 | 22.2 ± 1.1 | 21.6 ± 1.0 | 18.9 ± 1.1 | 19.8 ± 1.4 |
| CS-g-PCL | 24.5 ± 1.8 | 23.3 ± 1.2 | 23.5 ± 2.3 | 21.3 ± 0.9 | 19.3 ± 1.5 |
| *Shear rate [s$^{-1}$]* | | | | | |
| control | 60.3 ± 9.5 | 68.2 ± 9.4 | 41.4 ± 5.4 | 53.8 ± 6.5 | 56.9 ± 12.2 |
| PCL | 80.0 ± 14.5 | 75.0 ± 11.9 | 72.5 ± 12.7 | 83.6 ± 6.5 | 83.3 ± 9.3 |
| CS-g-PCL | 67.4 ± 8.4 | 89.9 ± 15.1 | 74.7 ± 11.1 | 100.5 ± 10.0 | 112.1 ± 26.0 |
| *Volumetric blood flow [pl/s]* | | | | | |
| control | 49.8 ± 6.2 | 59.5 ± 13.3 | 43.4 ± 10.3 | 39.6 ± 9.1 | 28.3 ± 11.2 |
| PCL | 56.1 ± 13.4 | 50.9 ± 10.6 | 44.4 ± 11.3 | 36.5 ± 7.0 | 41.4 ± 9.1 |
| CS-g-PCL | 69.1 ± 18.6 | 64.8 ± 8.5 | 57.0 ± 10.4 | 58.5 ± 5.1 | 50.4 ± 13.9 |

group with the CS-g-PCL-coated fiber mat. At day three the angiogenic response in the control group was comparatively high, but decreased over the evaluation period. Microvascular density showed constant data in the group of unmodified fiber mats. In contrast to the control group and the unmodified fiber mat, the CS-g-PCL coated fiber mat showed significantly increased angiogenesis 14 days after implantation (Table 3, Fig 3).

## Inflammatory response

The implantation did not significantly increase the number of rolling leukocytes in all groups. The rolling leukocytes were nearly constant in the control group and the CS-g-PCL group. The PCL group showed a gradual increase of rolling leukocytes over the evaluation period. The number of adherent leukocytes at the border zones of the implants decreased over the evaluation period in all groups however, there were no significant differences detectable between the groups over the observation period (Table 3, Fig 4).

**Table 3. Functional capillary density in cm/cm$^2$, number of rolling leukocytes in cells/min and number of adherent leukocytes in cells/mm$^2$ at the periphery of the control group scaffold (control), the unmodified PCL fiber mat group (PCL) and PCL fiber mat modified with CS-g-PCL group immediately (d0) and 3, 6, 10 and 14 days after implantation.** Values are expressed as means ± SEM.

| | Day 0 | Day 3 | Day 6 | Day 10 | Day 14 |
|---|---|---|---|---|---|
| *Functional capillary density [cm/cm$^2$]* | | | | | |
| control | 202.4 ± 11.1 | 177.4 ± 33.4 | 129.6 ± 33.2 | 144.4 ± 35.3 | 77.5 ± 31.6 |
| PCL | 85.4± 15.5 | 98.0 ± 13.9 | 91.8 ± 16.9 | 67.1 ± 8.4 | 89.0 ± 16.7 |
| CS-g-PCL | 97.9 ± 13.0 | 103.1 ± 16.6 | 93.5 ± 20.7 | 101.0 ± 11.3 | 148.2 ± 41.7 |
| *Number of rolling leukocytes [cells/min]* | | | | | |
| control | 3.6 ± 1.0 | 5.6 ± 2.1 | 5.0 ± 1.8 | 5.3 ± 2.1 | 3.8 ± 2.0 |
| PCL | 5.1 ± 1.9 | 5.3 ± 2.3 | 7.8 ± 2.4 | 12.0 ± 3.4 | 15.1 ± 4.7 |
| CS-g-PCL | 9.6 ± 4.3 | 11.0 ± 2.4 | 13.7 ± 3.1 | 11.2 ± 3.8 | 8.9 ± 3.8 |
| *Number of adherent leukocytes [cells/mm$^2$]* | | | | | |
| control | 358.4 ± 71.9 | 208.3 ± 44.2 | 337.9 ± 91.4 | 173.2 ± 73.4 | 100.7 ± 55.8 |
| PCL | 307.0 ± 97.8 | 276.8 ± 78.3 | 242.4 ± 122.9 | 50.0 ± 33.9 | 41.2 ± 29.8 |
| CS-g-PCL | 423.5 ± 129.2 | 360.8 ± 83.0 | 256.9 ± 71.9 | 80.1 ± 34.1 | 37.1 ± 24.4 |

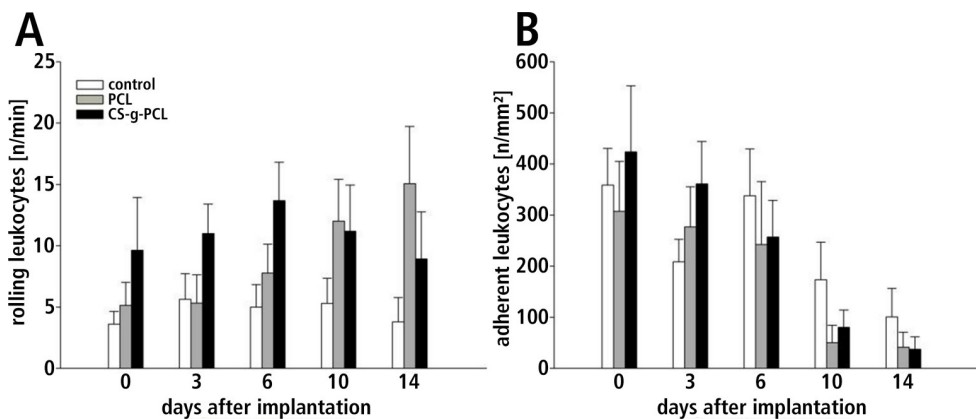

**Fig 3. Results of intravital microscopy. (A-C)** Intravital microscopic overview images of control **(A)**, PCL **(B)** and CS-g-PCL **(C)** at day 14. **(D)** Functional capillary density in cm/cm$^2$ in the border zones of the implants 0, 3, 6, 10 and 14 days after implantation. Means ± SEM; *p $<$ 0.05 vs. PCL and CS-g-PCL; +p $<$ 0.05 vs. PCL on the same day; °p $<$ 0.05 vs. CS-g-PCL on the preceding time point. Scale bars: 400 μm.

**Fig 4. Leukocyte-endothelium interaction.** Leukocyte-endothelium interaction at the periphery of the implants in post-capillary and collecting venules after implantation. **(A)** Number of rolling leukocytes 0, 3, 6, 10 and 14 days after implantation shown as number of cells/min. The implantation of different materials did not significantly increase the number of rolling leukocytes. **(B)** Number of adherent leukocytes 0, 3, 6, 10 and 14 days after implantation as number of cells/mm$^2$. The number of adherent leukocytes at the border zones of the implants decreased over the evaluation period.

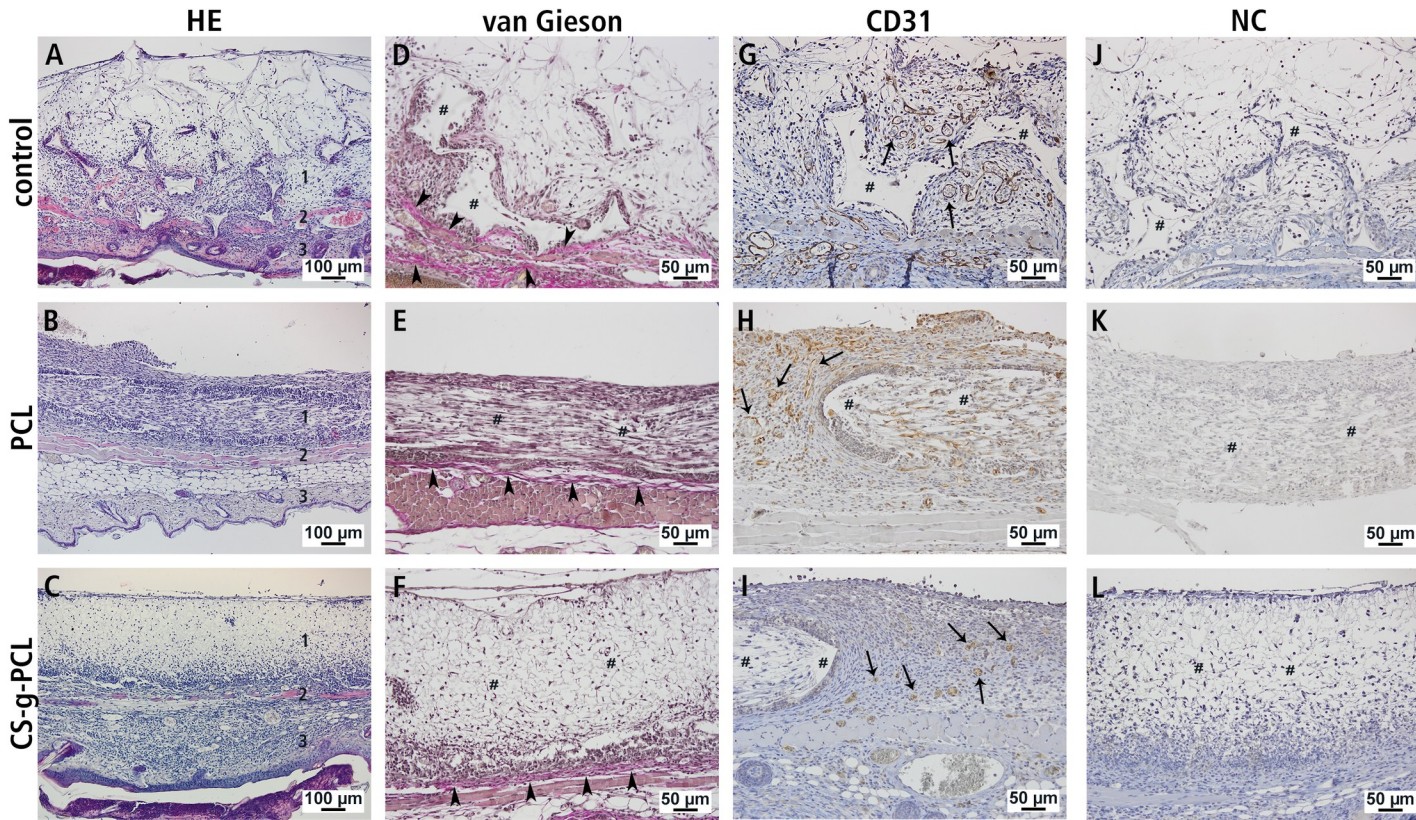

**Fig 5. HE staining, van Gieson staining and immunohistochemical detection of CD31.** Representative histological stainings **(A-I)** 14 days after implantation into the dorsal skinfold chamber of BALB/c mice. **(A-C)** HE staining **(1)** implant, **(2)** skin muscle, **(3)** subcutaneous tissue. **(D-F)** Collagen fibers were detected by van Gieson staining (arrowheads). **(G-I)** The presence of endothelial cells and accordingly enhanced functional capillary density was confirmed by immunohistochemical detection of CD31. **(J-L)** Negative controls. Areas of the implant are marked **(#)**, arrows denote vascular structures. Scale bars: 100 μm (A—C), 50 μm (D—L).

## Histology and immunohistochemistry

Hematoxylin-eosin-staining of sections from paraffin embedded specimen at day 14 showed marked differences in cell ingrowth depending on the implant material. The porous polymer patch was well infiltrated with cells throughout all regions of the scaffold (Fig 5A). The ingrowth of cells was also detectable in specimens with PCL fiber mats, but to a lesser extent. A cell-rich layer surrounding the PCL fiber mat was conspicuous in all specimens (Fig 5B). Also in the CS-g-PCL group a cellular margin was found around the implant, but less pronounced (Fig 5C). A comparable cell-rich layer around the implant was not detectable in the control group (Fig 5A).

Immunohistochemical examination with a CD31 antibody for detection of vascular structures at day 14 was applied to verify the results obtained with intravital fluorescence microscopy (Fig 5G, 5H and 5I). The granulation tissue in the border zones of the implanted materials was highly vascularized in all groups. However, endothelial cells were located in the surroundings of the PCL fiber mats (PCL and CS-g-PCL) but not within them, as compared to the control group where vascular structures were even detectable in the inside of implant structure. Negative controls for immunohistochemical detection of CD31, which showed no detectable staining, were performed by omitting the primary antibody (Fig 5J, 5K and 5L).

Collagen fibers were detectable in all groups by van Gieson staining. While in the control group collagen fibers were visible around the implant structure (Fig 5D), in the groups with

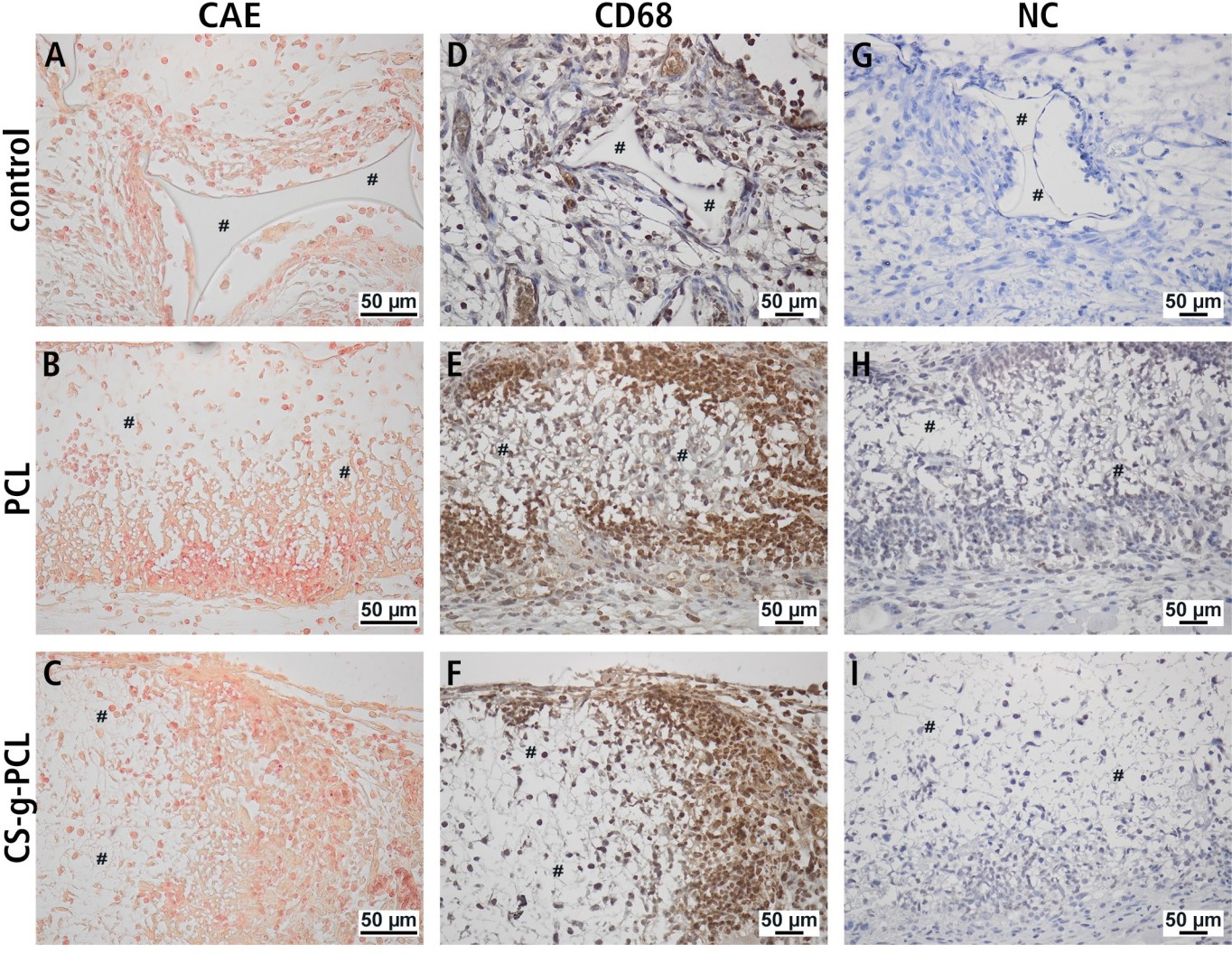

**Fig 6. Naphtol-AS-D-chloroacetate esterase staining and immunohistochemical detection of CD68.** Representative histological staining of neutrophilic granulocytes **(A-C)** 14 days after implantation into the dorsal skinfold chamber of BALB/c mice using naphtol-AS-D-chloroacetate-esterase (CAE) staining. Representative histological staining for detection of macrophages and monocytes **(D-I)** 14 days after implantation into the dorsal skinfold chamber of BALB/c mice. **(D-F)** Immunohistochemical detection of CD68 for detection of macrophages. **(G-I)** Negative controls. Areas of the implant are marked **(#)**.

the PCL fiber mats (PCL and CS-g-PCL) collagen fibers could be found just as a layer underneath the implant (Fig 5E and 5F).

The cellular composition of the cell-rich layer, which ensheathed the implants and the close vicinity of the implants, was further characterized by means of naphtol-AS-D-chloroacetate esterase staining and CD68 immunohistology (Fig 6). Chloroacetate-esterase positive

**Table 4. Results from quantification of immunofluorescent detection of immune cell markers.**

|  | control | PCL | CS-g-PCL |
|---|---|---|---|
| CSF1R | 0.070 ± 0.019 | 0.018 ± 0.005 | 0,029 ± 0,006 |
| CD86 | 0.086 ± 0,020 | 0.031 ± 0.012 | 0.035 ± 0.011 |
| CD11b | 0.112 ± 0.036 | 0.018 ± 0.006 | 0.061 ± 0.022 |
| CD3 | 0.072 ± 0.025 | 0.029 ± 0.015 | 0.020 ± 0.007 |

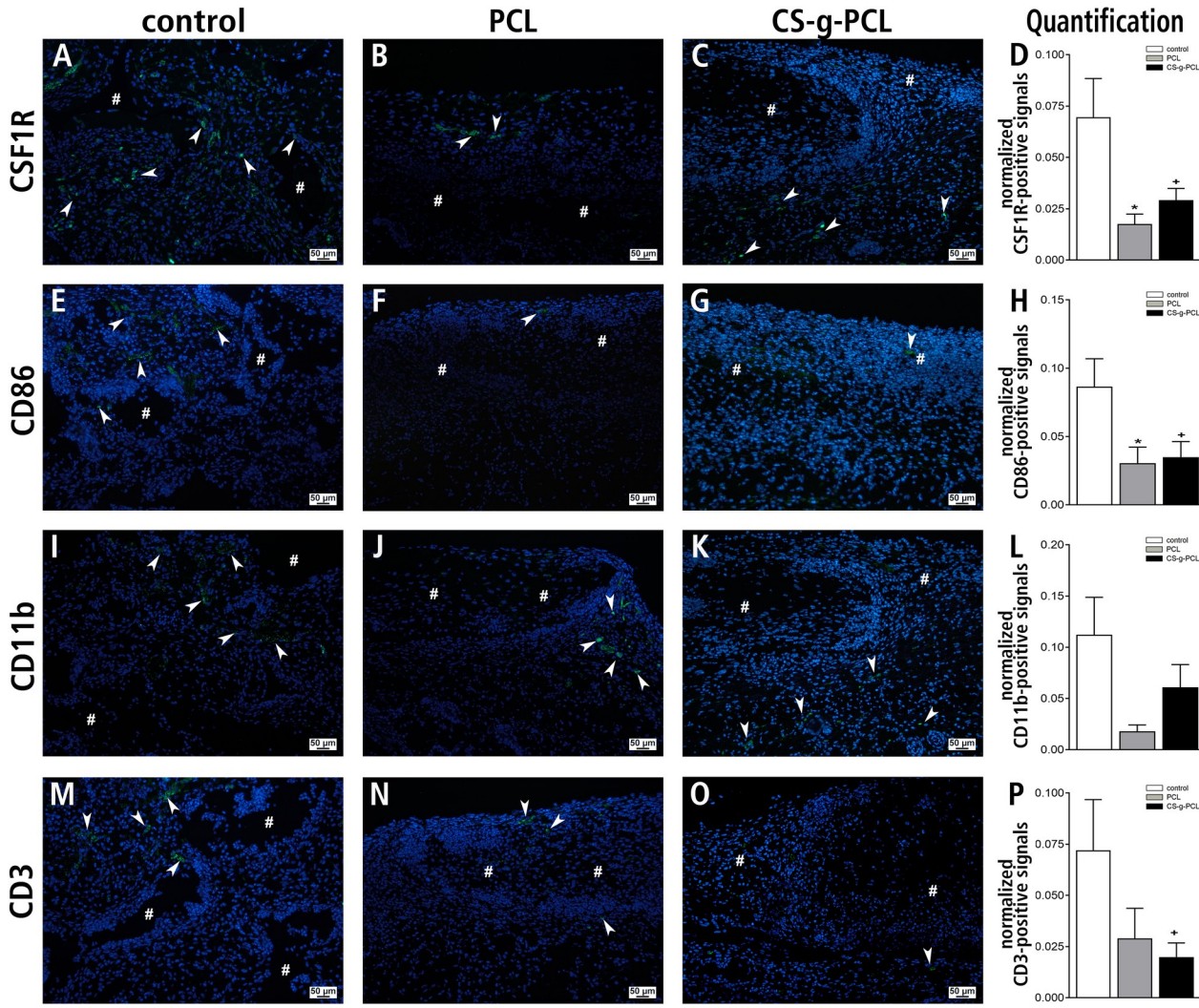

**Fig 7. Immunofluorescence detection of CSF1R, CD86, CD11b and CD3.** Representative histological stainings for detection of macrophages in general, macrophages with predominantly pro-inflammatory actions, monocytes and T-cells (all green fluorescence) **(A-C, E-G, I-K, M-O)** 14 days after implantation into the dorsal skinfold chamber of BALB/c mice. **(A-C)** Immunofluorescence detection of CSF1R for detection of macrophages in general. **(E-G)** Immunofluorescence detection of CD86 for detection of macrophages with predominantly pro-inflammatory actions. **(I-K)** Immunofluorescence detection of CD11b for detection of monocytes. **(M-O)** Immunofluorescence detection of CD3 for detection of T-cells. Nuclei were stained using DAPI (blue fluorescence). Scale bars: 50 μm. **(D,H,L,P)** Quantitative analysis of cell infiltration by immunofluorescent staining for CSF1R **(D)**, CD86 **(H)**, CD11b **(L)** and CD3 **(P)**. All data are expressed as the fluorescence intensity of the marker of interest normalized to the fluorescence intensity of DAPI staining. Values represent means ± SEM; *p < 0.05 vs. control and PCL; +p < 0.05 vs. control and CS-g-PCL.

neutrophilic granulocytes were detectable in all groups including the porous polymer patch. Although the neutrophilic granulocytes were detectable in all groups (Fig 6A, 6B and 6C), the PCL fiber mats groups, especially those with unmodified PCL (Fig 6E) showed a high infiltration of this cell type. Similar results were found by CD68 immunohistochemistry for detection of macrophages and monocytes (Fig 6D, 6E and 6F). Negative controls for immunohistochemical detection of CD68, which showed no detectable staining, were performed by omitting the primary antibody (Fig 6G, 6H and 6I).

Immunofluorescence staining of the marker proteins CSF1R, CD86, CD11b and CD3 were performed to further characterize the immune response around the implant (Table 4, Fig 7). By using these markers macrophages in general, macrophages with predominantly pro-

inflammatory actions, monocytes and T-cells were detectable nearly in all groups, but overall in a small extent. For all cell types, the control group showed the highest number of immune cells. We quantified and normalized the immunofluorescence signals and found significant differences for the respective immune cells. The detection of CSF1R and CD86 showed a significant lower amount of macrophages and macrophages with predominantly pro-inflammatory actions, respectively for both fiber mat groups. The detection of monocytes showed no statistically significant differences, although we found a clear lower number of this cell type in both fiber mat groups. The amount of the T-cell marker CD3 was clearly reduced in the PCL group and statistically significant reduced in the CS-g-PCL group.

## Discussion

One of the major challenges in the clinical application of implants is a fast and adequate vascularization of the construct after insertion to guarantee oxygen supply, nutrition of cells and removal of metabolites [10,33]. A lack or delay of angiogenesis and thus supply of nutrients is a common reason for inadequate healing after surgery. Furthermore the poor development of the microcirculation is known as a major problem in the integration of biomaterials [37]. As a consequence, implants should be designed to cause early angiogenesis resulting in a fast vascularization of the implant area and thereby promoting a fast and efficient regeneration. In consideration of these circumstances it is reasonable to test new implant materials with regard to vascularization and biocompatibility.

According to current knowledge, the main problem after rotator cuff tear repair is a high prevalence of re-rupture due to variable structural healing [38]. Until now there are only limited options for therapy; in particular there are no implants available, which bridge the defect site and facilitate the repair of tendon-bone-transitions. The critical step in creating a regenerated tendon-bone transition in rotator cuff tear repair is to achieve the functional integration of the implant material in the host tissue during the healing process. The absence of blood vessels in a repairing or metabolically active tissue may inhibit tissue repair; however, the rapid ingrowth of cells and tissues is indispensable regarding the transport of nutrients, waste and oxygen. To analyze microhemodynamic parameters on the muscle tissue over 14 days, the dorsal skinfold chamber is an ideal model. The implant material was tested in this soft tissue-model based on a muscular structure, which allows repeated quantitative assessment of implant vascularization in case of different applications, because the anatomical relationship of the transition between tendon and muscle is readjusted [33,39].

Electrospun fiber mats made of PCL were used for the experimental groups. The biocompatibility of PCL is well known, but its mechanical strength is not sufficient for load bearing applications [40]. Therefore, the electrospinning process was used to produce PCL fiber mats with aligned fibers for higher mechanical strength [25]. As mentioned above the porosity and pore size of the scaffold material is critical for ingrowth of capillaries, therefore based on the decrease in pore size and porosity caused by the modification a reduced vascularization could be anticipated for the CS-g-PCL coated samples. Furthermore chitosan is biocompatible and biodegradable, therefore being well known as a biological material which promotes the healing process of soft and hard tissues [41]. The immobilization of CS-g-PCL on top of electrospun PCL fibers by crystallization resulted in the modification of all fibers also within the material and not only on the surface like other techniques as e.g. plasma treatment [15].

The porous polymer patch, which was used as a control group in this study, has already been tested for biocompatibility [7] and showed an improved outcome in a level IV clinical study after augmentation surgeries [29]. Although the material characteristics of the porous polymer patch differ considerably from that of the experimental groups, this material was

selected as a control group as it has already been tested for the application of rotator cuff tear in animal studies and a small retrospective case study [7,29,42].

Before using implant materials for clinical applications, the knowledge about vascularization and biocompatibility is of great importance. Within the groups analyzed in our study we observed different patterns of vascularization. The control group showed a rapid vascularization and a high initial functional capillary density beginning with day three which declined at the next two days and further declined at the end of the observation period. A similar vascularization pattern can be observed during the healing of skin wounds. A histological analysis of this process showed an initially high density of small caliber capillaries. With the ongoing healing of the wound the number of vessels declined and vessel caliber increased [43]. We think that a comparable process accounts for the functional capillary density changes in our control group. Therefore we can conclude, that we observed a fast and complete ingrowth of vascularized tissue in the control scaffold. With respect to the CS-g-PCL group we think that angiogenesis and the growth of the granulation tissue is not that fast as it was observed in the control group and therefore there is still an increase in functional capillary density at the end of the observation period because tissue growth didn't stopped until that time point. The most important finding in this study was that CS-g-PCL coating improved the vascularization significantly compared to unmodified PCL fiber mats and the porous polymer patch in the long term. The significant increase in functional capillary density in PCL fiber mats with CS-g-PCL modification is a very promising result, especially as in different studies an enhanced vascularization was described as beneficial for the healing of rotator cuff tears. In an experimental study Harada et al. [44] used engineered cell sheets for the repair of resected infraspinatus tendons in rats. After 8 weeks the authors found a significant higher ultimate failure load in the cell sheet group compared the contralateral side which served as control without the implantation of a cell sheet. This enhanced mechanical stability was the result of an improved healing process which based on a higher vascularization of the repair site. An enhanced vascularization may also be beneficial for rotator cuff tear repair in human patients. In a pilot trial Zumstein et al. [45] used athroscopic repair of rotator cuff tears together with leukocyte- and platelet-rich fibrin. This treatment resulted in a higher vascularization six weeks after repair. Fealy et al. [46] also showed an enhanced vascularization immediately after rotator cuff repair by ultrasound imaging. It has to be mentioned that in both studies vascularization decreased at later time points and there were also no differences in clinical parameters at later time points. Nevertheless it seems to be feasible to assume that an enhanced vascularization is beneficial for the early stages of tendon healing after rotator cuff repair.

Recent studies demonstrated that vascularization of implants necessitates pores with a diameter of at least 300 μm [47]. Qualitative histologic examinations showed that the porous control polymer patch was well infiltrated with cells 14 days after implantation, because the spongy structure of this scaffold facilitated cell infiltration (but failed to provide mechanical stability). This finding was less apparent in the PCL fiber mats due to the denser structure despite their good biocompatibility. Furthermore, the histologic examinations showed capillaries in the inside of the control group in contrast to the PCL fiber mats, where the vascular structures were entirely limited to the outside of the implants. This result is attributable to the different structures of both materials. While the porous polymer control patch has a porous structure with an interconnected 3D network, which is favorable for cell ingrowth, the PCL fiber mats showed smaller structural spaces due to the aligned fibers [48]. Although the porous polymer patch improved biomechanical stability compared to controls without augmentation, it did not achieve the desired stability of native tendon in biomechanical examinations [7]. An optimal biomaterial should provide a porous texture, which is important for an adequate nutritional supply to the cells and cell-to-cell interactions. Furthermore, the porosity has to be

high enough to offer sufficient space for the sprouting of capillaries, cell proliferation, and integration into the surrounding tissue [47]. Therefore further developmental processes are necessary to generate PCL fiber mats that allow a sufficient cellular ingrowth and vascular supply.

The modification of PCL with chitosan resulted in an improved in vitro cytocompatibility compared to unmodified PCL [31]. The present study confirmed these data by in vivo data on leukocytes and histological data. In particular, the increase in rolling leukocytes in the PCL group is striking, while the values in the control group and the CS-g-PCL group remain nearly constant. Still, all measured levels for leukocytes are comparatively low in this study [28,35].

In terms of histology the unmodified PCL showed a prominent cellular margin around the implant. However, further characterization of this cell-rich layer around the implant showed only slightly increased numbers of macrophages and neutrophilic granulocytes compared to CS-g-PCL. The immunofluorescences even showed that significant fewer immune cells were detectable in both PCL and CS-g-PCL than in the control group. In terms of biocompatibility these results were promising especially for modified PCL fiber mats, as the porous polymer patch had already been classified as biocompatible in recent studies [32]. With respect to the short observation period, the collagen deposition did not allow any conclusion about a foreign body reaction, although slightly more collagen fibers were detectable in the PCL group.

In conclusion our study showed that electrospun PCL fiber mats coated with a CS-graft-PCL copolymer had a significant advantage for vascularization and biocompatibility of the implant in contrast to unmodified PCL fiber mats, even though the porosity and pore size was even smaller after modification with the graft-copolymer compared to unaltered PCL fiber mats. Clinically, these are important findings as they indicate that the modification with chitosan improves the formation of vascularized tissue and the ingrowth of cells. Furthermore the combination of enhanced vascularization and the reduction in immune cell activation at the later time points of our study points to an improved clinical outcome after rotator cuff tear repair. Further investigations should be directed toward enlargement of pore size within the fiber mats during their production to enhance subsequent cell ingrowth. In order to achieve this aim one approach may be to enlarge the remaining space between the aligned fibers, which can be achieved by increasing the fiber diameter [49].

## Supporting information

**S1 File. ARRIVE (Animal Research: Reporting of In Vivo Experiments) guidelines checklist.**
(PDF)

## Acknowledgments

We acknowledge the excellent technical assistance of Stefanie Rausch. This research project has been supported by the German Research foundation (DFG), research unit FOR 2180 "Gradierte Implantate für Sehnen-Knochen-Verbindungen".

## Author Contributions

**Conceptualization:** Andreas Kampmann.

**Formal analysis:** Nils Ludwig.

**Funding acquisition:** Andrea Hoffmann, Andreas Kampmann.

**Investigation:** Sarah Gniesmer, Anna Lena Hoheisel.

**Methodology:** Sarah Gniesmer, Ralph Brehm, Dominik de Cassan, Anna Lena Hoheisel, Elmar Willbold.

**Project administration:** Andrea Hoffmann, Andreas Kampmann.

**Resources:** Andreas Kampmann.

**Supervision:** Ralph Brehm, Andreas Kampmann.

**Validation:** Andreas Kampmann.

**Visualization:** Nils Ludwig.

**Writing – original draft:** Sarah Gniesmer.

**Writing – review & editing:** Sarah Gniesmer, Ralph Brehm, Andrea Hoffmann, Dominik de Cassan, Henning Menzel, Anna Lena Hoheisel, Birgit Glasmacher, Elmar Willbold, Janin Reifenrath, Nils Ludwig, Ruediger Zimmerer, Frank Tavassol, Nils-Claudius Gellrich, Andreas Kampmann.

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
