## [Decision Letter · Decision Letter 0]

23 Sep 2019

PONE-D-19-22863

Vascularization and biocompatibility of poly(ε-caprolactone) fiber mats for rotator cuff tear repair

PLOS ONE

Dear Dr. Kampmann,

Thank you for submitting your manuscript to PLOS ONE. After careful consideration, we feel that it has merit but does not fully meet PLOS ONE’s publication criteria as it currently stands. Therefore, we invite you to submit a revised version of the manuscript that addresses the points raised during the review process.

We would appreciate receiving your revised manuscript by Nov 07 2019 11:59PM. To enhance the reproducibility of your results, we recommend that if applicable you deposit your laboratory protocols in protocols.io, where a protocol can be assigned its own identifier (DOI) such that it can be cited independently in the future. For instructions see: http://journals.plos.org/plosone/s/submission-guidelines#loc-laboratory-protocols

We look forward to receiving your revised manuscript.

Kind regards,

Feng Zhao

Academic Editor

PLOS ONE

2. Please complete and submit a copy of the ARRIVE Guidelines checklist, a document that aims to improve experimental reporting and reproducibility of animal studies for purposes of post-publication data analysis and reproducibility: https://www.nc3rs.org.uk/arrive-guidelines. Please include your completed checklist as a Supporting Information file. Note that if your paper is accepted for publication, this checklist will be published as part of your article.

Additionally please include the source of the mice used in your study and the minimal information necessary to prepare the electrospun polycaprolactone fiber mats (we note that you have provided a reference but we would recommend including some basic information in this submission).

**Comments to the Author**

1. Is the manuscript technically sound, and do the data support the conclusions?

Reviewer #1: Yes

Reviewer #2: Partly

2. Has the statistical analysis been performed appropriately and rigorously? 

Reviewer #1: Yes

Reviewer #2: No

3. Have the authors made all data underlying the findings in their manuscript fully available?

Reviewer #1: Yes

Reviewer #2: Yes

4. Is the manuscript presented in an intelligible fashion and written in standard English?

Reviewer #1: Yes

Reviewer #2: Yes

5. Review Comments to the Author

Reviewer #1: The present study shows vascularization of Chitosan grafted PCL scaffolds in dorsal skinfold chamber model. The authors have showed that chitosan grafted PCL significantly improved vascularization compared to unmodified PCL fiber mats and the porous Biomerix patch after 2 weeks of implantation, which can be beneficial for improved tendon repair after tendon injury during rotator cuff tear. Although technically sound, the manuscript requires few corrections as mentioned below.

Line: 196 Briefly explain the electrospinning parameters in this section as ref 20 might not be available for everyone.

Line 266: briefly mention the method for analyzing: (1) functional capillary density, (2) Number of rolling as well as adherent leukocytes

Description regarding Fig 5 (J-L) is missing.

Minor comments:

Line 422: It Should be “(Figure 6 D, E, F)” instead of (Figure 6 A, B, C).

Figure 6 (G, H, I): explain what do you mean by negative control. Is it H&E staining?

Figure 7: Please mark the implants (i.e. using #).

Reviewer #2: The animal model used lacks relevance in the application specific environment of the proposed treatment. The insight provided by the animal model is useful from a basic standpoint ( of immunogenicity of materials and subsequent vascularization), but cannot be directly linked to rotator cuff injuries and healing progression. The manuscript should be completely revised, centered around the use of the animal model for engineering materials to improve their vascularization and immunogenicity in general, without a specified application.

It would be useful for the authors to quantify the results in Fig.7.

Fig. 3 needs more explanation. For instance, the control material has a high initial capillary density at days 3 and 10, then decreases. Conversely, the density for the CS-g-PCL material is low initially then suddenly increases. What is the explanation for these two different vascularization patterns?

The authors conclude that CS-g-PCL has improved capillary function, based only on one evaluation day out of the four. This would be more convincing if the entire time course was considered.

It appears that no statistical test, or the lack of statistical significance is prevelant for the histological data. The authors state that the CS-g-PCL material has reduced immunogenicity, yet there are no statistical differences shown in the graphs or results.

Furthermore, the author should explain the reasoning for the post hoc test chosen, among others.

The discussion section needs to be better organized. It its rather hard to follow, and makes loose connections between the results and speculated claims.

---

## [Author Response · Author response to Decision Letter 0]

6 Nov 2019

Reply to the major comments of reviewer 1:

1. A brief description of the electrospinning parameters has been added to the Materials and methods section of our manuscript. A further description seems not to be necessary as reference 20 is an open access publication. Unfortunately reference 20 was not cited correctly in the reference list, we corrected this, so that the information is available for every reader. (see lines 194-200)

2. We added a brief description of the intravital microscopy data analysis to the Materials and Methods section. (see lines 272-286)

3. Figure 5 J-L are negative controls for the immunhistochemical detection of CD31, we added a short description to the respective paragraph of the results section. (see lines 424 and 425)

Reply to the minor comments of reviewer 1:

1. Thank you very much or this hint. We corrected the description of figure 6 accordingly. (see lines 443, 444 and 446)

2. As described in the Material and methods section, we performed negative controls be omitting the primary antibody, leaving the rest of the used protocol unchanged. This information is already present in the Materials and methods section of the manuscript, but we also added a short description to the respective paragraph of the results section. (see lines 446-448)

3. As suggested by the reviewer we marked the implants in the microscopic pictures presented in Figure 7 using the same scheme as in Figures 3, 5 and 6. (see updated Figure 7)

Reply to the comments of reviewer 2:

1. Our study describes one step in the development process of a new implant for the augmentation of rotator cuff tears. Thereby our research is focused on the vascularization of the implant because this is an important fact, which has considerable impact on the success of the implant. Therefore we don’t agree with the reviewer that our results lack relevance in respect to rotator cuff tears. In different studies an enhanced vascularization was described as beneficial for the healing of rotator cuff tears. In an experimental study Harada et al. [1] used engineered cell sheets for the repair of resected infraspinatus tendons in rats. After 8 weeks the authors found a significant higher ultimate failure load in the cell sheet group compared the contralateral side which served as control without the implantation of a cell sheet. This enhanced mechanical stability was the result of an improved healing process which based on a higher vascularization of the repair site. An enhanced vascularization may also be beneficial for rotator cuff tear repair in human patients. For example Funakoshi et al. [2] showed an increased vascularization of the tendons one or two months after rotator cuff tear repair, which decreased until three months after surgery. In a pilot trial Zumstein et al. [3] used athroscopic repair of rotator cuff tears together with leukocyte- and platelet-rich fibrin. This treatment resulted in a higher vascularization six weeks after repair. Fealy et al. [4] also showed an enhanced vascularization immediately after rotator cuff repair by ultrasound imaging. It has to be mentioned that in both studies vascularization decreased at later time points and there were also no differences in clinical parameters at later time points. Nevertheless it seems to be feasible to assume that an enhanced vascularization is beneficial for the early stages of tendon healing after rotator cuff repair. Therefore we decided to test possible implants for rotator cuff tear repair first in a model that allows the evaluation of vascularization. Implants that were beneficial for vascularization are possible candidates for further research in an animal model of rotator cuff repair. To further enhance our manuscript, we added parts of this argumentation to the discussion section of our manuscript.

2. As suggested by the reviewer we quantified the results in Figure 7 using the software cellSens Dimensions 1.14 (Olympus Deutschland GmbH, Hamburg, Germany). The results of the quantification were added as further information to figure 7. Please refer to our answer to comment 5 for a detailed reply.

3. The different vascularization patterns depicted in Figure 3 can be explained by a different extent of vascular remodeling. The control group showed a rapid vascularization and a high initial functional capillary density beginning with day three. The functional capillary density in the control group showed a decline for the next two days, which was not statistically significant and further declined at the end of the observation period (day 14, statistically significant). A similar vascularization pattern could be observed during the healing of skin wounds. A histological analysis of this process showed an initially high density of small caliber capillaries. With the ongoing healing of the wound the number of vessels declined and vessel caliber increased. [5] We think that a comparable process accounts for the functional capillary density changes in our control group. Therefore we can conclude, that we observed a fast and complete ingrowth of vascularized tissue in the Biomerix scaffold. With respect to the CS-g-PCL group we think that angiogenesis and the growth of the granulation tissue is not that fast as it was observed in the control group and therefore there is still an increase in functional capillary density at the end of the observation period because tissue growth didn’t stopped until that time point.

4. Taken our answer to comment 3 and with respect to the possible clinical application of electrospun fiber mats as scaffolds for the repair of rotator cuff tears we think that the observed changes in functional capillary density are a tolerable, if not desirable feature of such a scaffold. As stated in the answer to comment 1 an enhanced vascularization was described as beneficial for the healing of rotator cuff tears. We therefore think that especially CS-g-PCL fiber mats are well suited for rotator cuff tear repair as they combine a high vascularization with a good biocompatibility.

5. As already stated in our answer to comment 2, we quantified the results of the immunofluorescence detection of macrophages (CD68, CSF1R, CD11b) and T-cells (CD3). Although there were no statistically significant differences for CD11b, the other results supported our hypothesis as they showed a significant lower accumulation of macrophages and T-cells after implantation of PCL and CS-g-PCL implants. We added this new information to the results and discussion section of our manuscript. (see lines 318-324, lines 463-469, lines 481-484, the new Table 4, the new Fig 7 and the related parts of the discussion)

6. We used the Student-Newman-Keuls post-hoc test for differences in means. Once ANOVA gave us statistically significant results, we performed this test to assess which specific pairs of means are different, based on the studenized range distribution. Student-Newman-Keuls is hereby a powerful test and less conservative as other pot hoc tests such as Tukey's range test. We therefore decided to stick to the Student-Newman-Keuls method throughout the whole study.

7. To further enhance the discussion of our manuscript we reorganized the discussion section, thereby adding a discussion of the new information which was added according to the reviewer’s suggestions.

References:

1. Harada Y, Mifune Y, Inui A, Sakata R, Muto T, Takase F, Ueda Y, Kataoka T, Kokubu T, Kuroda R, Kurosaka M. Rotator cuff repair using cell sheets derived from human rotator cuff in a rat model. J Orthop Res. 2017 Feb;35(2):289-296.

2. Funakoshi T, Iwasaki N, Kamishima T, Nishida M, Ito Y, Nishida K, Motomiya M, Suenaga N, Minami A. In vivo vascularity alterations in repaired rotator cuffs determined by contrast-enhanced ultrasound. Am J Sports Med. 2011 Dec;39(12):2640-6.

3. Zumstein MA, Rumian A, Lesbats V, Schaer M, Boileau P. Increased vascularization during early healing after biologic augmentation in repair of chronic rotator cuff tears using autologous leukocyte- and platelet-rich fibrin (L-PRF): a prospective randomized controlled pilot trial. J Shoulder Elbow Surg. 2014 Jan;23(1):3-12.

4. Fealy S, Adler RS, Drakos MC, Kelly AM, Allen AA, Cordasco FA, Warren RF, O'Brien SJ. Patterns of vascular and anatomical response after rotator cuff repair. Am J Sports Med. 2006 Jan;34(1):120-7.

5. Johnson KE, Wilgus TA. Vascular Endothelial Growth Factor and Angiogenesis in the Regulation of Cutaneous Wound Repair. Adv Wound Care (New Rochelle). 2014 Oct 1;3(10):647-661.

---

## [Decision Letter · Decision Letter 1]

17 Dec 2019

PONE-D-19-22863R1

Vascularization and biocompatibility of poly(ε-caprolactone) fiber mats for rotator cuff tear repair

PLOS ONE

Dear Dr. Kampmann,

Thank you for submitting your manuscript to PLOS ONE. After careful consideration, we feel that it has merit but does not fully meet PLOS ONE’s publication criteria as it currently stands. Therefore, we invite you to submit a revised version of the manuscript that addresses the points raised during the review process.

We would appreciate receiving your revised manuscript by Jan 31 2020 11:59PM. To enhance the reproducibility of your results, we recommend that if applicable you deposit your laboratory protocols in protocols.io, where a protocol can be assigned its own identifier (DOI) such that it can be cited independently in the future. For instructions see: http://journals.plos.org/plosone/s/submission-guidelines#loc-laboratory-protocols

We look forward to receiving your revised manuscript.

Kind regards,

Feng Zhao

Academic Editor

PLOS ONE

**Comments to the Author**

1. If the authors have adequately addressed your comments raised in a previous round of review and you feel that this manuscript is now acceptable for publication, you may indicate that here to bypass the “Comments to the Author” section, enter your conflict of interest statement in the “Confidential to Editor” section, and submit your "Accept" recommendation.

Reviewer #1: All comments have been addressed

Reviewer #3: (No Response)

2. Is the manuscript technically sound, and do the data support the conclusions?

Reviewer #1: Yes

Reviewer #3: Yes

3. Has the statistical analysis been performed appropriately and rigorously? 

Reviewer #1: Yes

Reviewer #3: Yes

4. Have the authors made all data underlying the findings in their manuscript fully available?

Reviewer #1: Yes

Reviewer #3: Yes

5. Is the manuscript presented in an intelligible fashion and written in standard English?

Reviewer #1: Yes

Reviewer #3: Yes

6. Review Comments to the Author

Reviewer #1: (No Response)

Reviewer #3: Authors used the dorsal skinfold chamber model to test the pro-vascularization property of electrospun PCL-based fiber mats. The commercially available porous polyurethane patch (Biomerix™ RCR patch) was used as the control. Both pristine and chitosan-graft-PCL coated electrospun PCL fiber mats were tested. In the chitosan-graft-PCL group at Day 14 vascularization was significantly enhanced, and a reduced activation of immune cells was observed. It is claimed that the CS-g-PCL may benefit the healing of rotator cuff tears by improving the ingrowth of capillaries into electrospun PCL scaffolds and reducing the immune cell activation at the later stage after the implantation. The study was carefully and systematically carried out. However, there are still some minor issues need to be addressed before official publication.

1. The biggest concern is after going through the whole manuscript, the reviewer feels hard to understand why the PCL fiber is specifically beneficial to rotator cuff tear repair. Note that PCL electrospun fiber has been promising for vasculature, bladder, bone, cartilage, and skin tissue engineering. In addition, vascularization and immune response are common issues in all types of tissue engineering. It feels like if rotator cuff tear repair were replaced by vascular tissue repair, the data would still be OK to use. As a result, one suggestion is that in the intro section, detailed background on the use of Biomerix™ RCR patch and PCL specifically for rotator cuff tear or tendon repair should be given, for example in the second paragraph, at Line 180ish.

2. In the paper, authors claim “that the CS-g-PCL improves the ingrowth of capillaries into electrospun PCL scaffolds”. However, the data actually indicate the PCL mats improve the vascularization around not in the implant. Please use more accurate expressions.

3. Experiment section on evaluation of porosity and pore diameter should be re-written. Brief description is needed instead of simply citing other references.

4. Line 251, what do you mean by “in a circular area of 12 mm”? Do you mean “with a diameter of 12 mm?”

5. Table 1, be careful about the significant figures. 0.6594 μm seems meaningless compared with 1.6 μm. Also, be consistent about the significant figures.

---

## [Author Response · Author response to Decision Letter 1]

19 Dec 2019

Reply to the minor comments of reviewer 3:

1. We thank the reviewer for this suggestion. To better illustrate the reasons why electrospun PCL is well suited for rotator cuff tear repair und to show why we used the specific configuration of the PCL fiber mats, we added additional information, concerning the use of PCL in different tissue engineering application and especially for the repair of rotator cuff tears. Implants for rotator cuff tear repair should have a positive impact on tendon healing by supporting cellular infiltration and guiding the regeneration of an organized tendon structure. By fabrication of fiber mats with directed or undirected fiber orientation they can be adopted to the microarchitecture of the different sections of the native tendon. The specific design of the fiber mat in turn stimulates the regeneration of organized tendon structures. In a previous study we analyzed electrospun PCL fiber mats with undirected fiber orientation, that were intended to mimic the specific fiber orientation at the tendon-bone transition in the femur chamber in rats [Gniesmer et al. 2019]. The current study was designed to examine fiber mats with directed fiber structures, which simulate the tendon-muscle transition. These electrospun PCL fiber mats are intended to form the lead structure for the transition zone of the rotator cuff. (see lines 180-195)

2. We apologies for the confusion caused. The expression “ingrowth of capillaries” was used in the abstract of the manuscript and the conclusion at the end of the discussion section. For a clear discussion of our results we changed the expression to “formation of vascularized tissue and the ingrowth of cells” (see lines 137 and 624)

3. We updated the Materials and methods section and added a brief description how the evaluation of porosity and pore diameter was done. (see lines 227-233)

4. The reviewer is right at this point; by mistake we omitted essential information about the preparation of the observation window. We corrected this mistake in the revised version of the manuscript. (see lines 270 and 271)

5. In the revised version of the manuscript we tried to harmonize the number of digits after the decimal point where possible. In Table 1, 2 and 3 we used one decimal place. In Table 4 the use of three decimal places was necessary and justified by our data. We also corrected wrong use of decimal point and decimal comma in all tables. (see Table 1, 3 and 4)

Reference:

Gniesmer S, Brehm R, Hoffmann A, de Cassan D, Menzel H, Hoheisel A-L, et al. In vivo analysis of vascularization and biocompatibility of electrospun polycaprolactone fiber mats in the rat femur chamber. J Tissue Eng Regen Med. 2019 Apr 26;0(ja). https://doi.org/10.1002/term.2868

---

## [Editor Report · Decision Letter 2]

23 Dec 2019

Vascularization and biocompatibility of poly(ε-caprolactone) fiber mats for rotator cuff tear repair

PONE-D-19-22863R2

Dear Dr. Kampmann,

We are pleased to inform you that your manuscript has been judged scientifically suitable for publication and will be formally accepted for publication once it complies with all outstanding technical requirements.

With kind regards,

Feng Zhao

Academic Editor

PLOS ONE

---

## [Editor Report · Acceptance letter]

3 Jan 2020

PONE-D-19-22863R2 

Vascularization and biocompatibility of poly(ε-caprolactone) fiber mats for rotator cuff tear repair 

Dear Dr. Kampmann:

I am pleased to inform you that your manuscript has been deemed suitable for publication in PLOS ONE. Congratulations! Your manuscript is now with our production department. 

With kind regards,

on behalf of

Dr. Feng Zhao 

Academic Editor

PLOS ONE